# Pre- and Post-Occupancy Evaluation of Resident Motivations for and Experiences of Establishing a Home in a Low-Carbon Development

**Jessica K. Breadsell** *[ID], **Joshua J. Byrne** and **Gregory M. Morrison**

Curtin University Sustainability Policy Institute, School of Design and the Built Environment, Curtin University, Bentley 6102, Australia

*** Correspondence: Jessica.breadsell@curtin.edu.au

**Abstract:** There is some understanding of how an individual's daily practices consume resources in the home, but the home as a space itself and peoples' relationships to it remain an interesting research area. In this paper, residents of an Australian low-carbon development (LCD) are studied in order to discover the expectations and motivations driving them to move to their new home, the emotional landscape of the home, and their subsequent experiences living in an LCD. This exploration through mixed methods and a post-occupancy evaluation enables a longitudinal empirical study of the motivations, perceptions, expectations and experiences of an LCD residence. This study aims to further conceptualize the social understanding of a home and what people consider when moving into an LCD, along with the post-occupancy experiences that are important for establishing LCDs in the future. The results show that a home is associated with being a place of community, sustainability, safety and comfort, as well as a place that incorporates aesthetically pleasing features. The motivation for residents moving into an LCD is to have housing stability, live the life they want (including performing sustainable practices) and enjoy the attractive design of the LCD. The user experiences of living in an LCD include unexpected design influences on daily practices and an appreciation of the community atmosphere created. The strong sense of community and the self-reported thermally comfortable homes met residents' expectations post-occupancy. This research is of interest to academics in the low-carbon and social science sectors, real-estate agents and property developers, as it provides insight into motivations and expectations of low-carbon dwelling residents.

**Keywords:** low carbon development; user experience; post-occupancy evaluation; home perceptions; Australia

## 1. Introduction

Given the surge of population towards 9 billion by 2050 and the concomitant rapid urbanization, cities around the world will have to accommodate additional dwellings while adapting to the climatic and spatial challenges facing them. The prevalence of low-carbon or zero energy homes is being driven by an international policy push to standardize these dwellings over the next few decades in the interests of climate and spatial issues [1]. Australia has one of the highest levels of greenhouse gas emissions per capita, with the main source being energy for electricity in homes [2,3]. This is being driven by high energy requirements in buildings and homes that are not thermally comfortable and have high mechanical heating and cooling uses by residents [4]. In Australia, building codes now require houses to meet minimum energy-efficiency requirements around orientation for natural ventilation and sunlight levels and building materials used [5], and there are ongoing efforts by both industry and government to see these regulations increased [6–8]. There are a number of projects around the country

that showcase how (beyond compliance) energy efficiency and other sustainability initiatives can be incorporated into highly livable developments in Australia. Low- carbon developments (LCD) such as Christie Walk, the Commons, Bowden, Central Park and the WGV development provide working examples at different scales [9]. These developments have focused primarily on the physical design of the developments, but questions remain around how residents perceive LCDs, including whether the design of the home is a major concern for residents, what drives these perceptions and expectations and what levels of post-occupancy satisfaction exist. If Australia and the rest of the world continue building low-carbon houses, then the motivations behind people moving into these homes are important to understand. This paper will explore the resident attraction to a 2.2-ha medium-density residential LCD in Western Australia, called WGV, and residents' post occupancy experiences of their new homes. Perth, the capital of Western Australia, has one of the highest uptakes of solar PV systems in the country, with 25% of all residential homes possessing rooftop systems [10] and government policies continuing to promote higher densities. As such, Perth provides a good case study for assessing these changes in the housing market [11].

## 2. Relevant Literature

Previous studies examining housing preference in Perth, Australia generally and globally [12–15] have used a stated preference market-choices methodology [14,16], as distinct from what people have actually chosen when purchasing a home. There is often a large variation in what people would state they would buy versus what they actually do buy [16]. Therefore, less emphasis can be placed on findings from stated market preference studies [16,17] compared to studies with actual market-preference methods. Stated market preference was originally used to examine the difference between preferences revealed in surveys or experiments and those observed in actual behavior. Their use has evolved from economic theories in 1953, to the study of transportation preferences in the 1980s and 1990s to environmental preferences more recently [18]. Stated housing preference studies in Perth, Australia generally and globally have concluded that people are most influenced by the type of housing, as well as its affordability, location and size, when choosing where to live [12–15]. The Australian studies found that people prefer to own their own home, typically a large, detached house near a city center. This type of dwelling provides opportunities for self-expression, privacy and autonomy, as well as offering space to relax with less interaction with or interference from neighbors [14,16].

Literature examining housing preferences for low-energy or low-carbon homes has been less prevalent in Australia than in Europe. Post-occupancy evaluations of houses has centered on studies of Passive House residents, concluding that residents are generally more thermally comfortable in their dwellings in winter and appreciate the improved indoor air quality [19]. Post-occupancy studies of low-energy buildings in Australia have focused on occupant comfort and interaction with technologies in the dwellings [1,4,20,21]. These studies have found that many occupants of LCDs have little to no experience with the new technologies and how to effectively use them to remain comfortable in their homes [22]. However, individual user experiences are highly personal, and the many reasons that motivate people to move into an LCD, including health and well-being, lifestyle, environmental beliefs or simply price and location, should be acknowledged by builders, real-estate agents and policy makers [19,23–25]. Purpose-built low-energy houses in the UK were found to economically empower low-income residents through reducing energy bill stress and allowing income to be spent instead on family time together [25].

The absence of empirical evidence documenting residents' perceptions of these low-energy and low-carbon homes, particularly in regards to climate variability, limits the ability of policy makers and designers to understand residents' lives at home [20,21]. The work conducted by [26] highlighted this, and showed that a sustainable home is more than just an energy-efficient building—it must encompass a holistic view of the economic, environmental and social aspects of residents' lives. Evaluations by researchers should focus on users' reasons for choosing to live in energy-efficient buildings, to be able

to give input on how to market energy-efficient buildings [23]. This paper contributes to addressing this knowledge gap.

This research also investigates the meanings and emotional landscapes attributed to homes by LCD residents. There are different perspectives of the home advocated through the literature that have provided succinct overviews of the research undertaken, and a selection of these aspects have been compiled in Table 1 [27–32]. As [33] states, a home is a feeling, a sense of comfort or belonging, and not necessarily a location. A person can live in a house and not feel at home [31]. The physical attributes a home provides (security, a place to raise a family, a place to perform activities and ownership) were the primary attributes a home was given in the literature pre-2000. In [29], a summary of the literature view on home perceptions was undertaken to clarify the inclusion of other aspects in the meaning of "home" to reflect the social and personal space it provides. This line of reasoning was continued in [34,35], concluding that people desire housing for its provision of both material security and an emotionally stable environment, especially individuals who have faced homelessness and housing instability. This paper investigates the aspects of a home that reflect the values of a person moving into an LCD, and uses this prior literature for context.

**Table 1.** A selection of aspects of the meaning of home identified in the research literature (author compiled).

| Aspects of Home | References |
| --- | --- |
| Security and safety | [28,34–41] |
| A place to raise children and have relationships over generations | [28,29,32,34,36–38,40–44] |
| An asset, a place to own | [35–38,40] |
| A place to spend time and undertake activities | [32,37,40,41,43–45] |
| A place for privacy, a haven and being away from the world | [28,29,34,36–38,44,46] |
| A place to do what you like in, a sense of control | [28,32,34,36–40,42,43,45,46] |
| A reflection of one's ideas, values, identity and emotional landscape | [28,29,32,34,35,37,39,40,42,43,45–47] |
| A site of consistency and permanence | [28,29,32,35,37,38,40,41,45] |
| A site of engagement with community/neighbors | [29,34,39,41] |

## 3. Methods

This research is based on a pre- and post-occupancy evaluation of an LCD. Post-occupancy evaluation is an established method of studying occupants of buildings for feedback and/or through measurements of building performance [19,24,48]. The occupants of low-carbon and similar homes (passive houses, low-energy houses, zero-energy houses) have been described as a special segment of the population with specific lifestyles, behaviors and practices and views. This is due to being early adopters of new technology, housing and community designs that are not standard for the rest of the population. The way these residents interact with these features can be studied to improve the uptake and acceptance of LCDs. With an increase in low-carbon homes around Australia, the study of these residents is vital in understanding how these buildings are integrated into society in the future. Therefore, this research will center on an LCD in Perth, Western Australia, called WGV [49].

In this paper, "LCD" refers to a group of households that form part of a development with design performance requirements beyond the Australian National Construction Code (e.g., 7+ star NatHERS thermal performance) and inclusion of a solar PV system. This standard can be met through natural ventilation, orientation of the dwelling to take advantage of the sun, shading through awnings and verandas and building materials used, including double glazing. The WGV development studied consists of multiple dwelling types and will comprise approximately 80 dwellings when completed, including multi-story dwellings flexibly connected to a broader energy system so that a home is no longer a single dwelling but part of a system (as discussed in [32]). The first residents began moving

in to WGV in mid-2017. The homes were designed for a Mediterranean climate, with sustainability features including a passive solar design that allows airflow and sunlight levels (solar gain) to assist the regulation of international temperature. The average outdoor temperature is between 10 °C and 27.3 °C annually [50].

A cohort study of 14 residents inhabiting 13 homes (n = 14) was undertaken, with data collected both pre- and post-occupancy in the LCD. The residents studied had moved into a variety of dwelling typologies. One cohort (five residents studied) was Sustainable Housing for Artists and Creatives (SHAC), who were leasing apartments and two studio spaces from a local social housing provider, with rental payment concessions received from the Australian Government. Another cohort (six residents studied) were owner-occupiers of apartments sold at market rates in a commercial development called Evermore. The third cohort (three residents studied) were owner-occupiers of two semi-detached units, while the final resident studied was an owner-occupier of a stand-alone (detached) house. Three households across two cohorts had previously had sustainability features in their homes.

Mixed methods were employed pre- and post-occupancy for data collection [51,52]. The data collection methods focused on the themes of energy, water, waste, food, transport, social network practices and residents' expectations and motivations for moving into WGV. This paper focuses on concepts surrounding moving into WGV, such as the expectations and motivations for the move, definitions of home and how they changed, how the residents were experiencing living in WGV and community experience. The residents' practices concerning energy, water, recycling, shopping, transport and food will be discussed in forthcoming papers.

Residents self-selected through an open invitation sent to those who had already purchased property in the LCD or were intending to become a tenant through SHAC (n = 27). An original sample size of 16 individuals in 15 dwellings were part of the pre-occupancy data collection; however, one household decided to rent out their apartment in WGV, and another removed themselves from the study. Their results are not included in this paper. Pre-occupancy data collection was conducted between April and June 2017 for SHAC residents, and between December 2017 and March 2018 for Evermore and single-house residents. Post-occupancy data collection was conducted between December 2018 and March 2019 for all residents. The long period of time for data collection pre-occupancy was intended to allow for a greater sample size of residents to self-select. However, there was a post-occupancy bias towards those in SHAC or Evermore, due to the requirement that the resident reside in the LCD during 2018, to allow for post-occupancy data collection within the research time constraints.

A structured interview explored the occupants' motivations and experiences surrounding the move to the LCD, while text probes, hygiene and transport diaries provided contextual experience data. The interviews (questions in the semi-structured interview asked residents how they kept warm and cool, the routines they went through each day and how their lives had changed since moving to the LCD) were for approximately one hour and were undertaken in the residents' pre- and post-occupancy accommodations, except for one which was conducted at an independent venue. In households with multiple adults, only those moving into the LCD were interviewed. Children, including those over 18 and still living at home, were not interviewed due to uncertain circumstances surrounding their residency arrangements once their parents moved into the LCD.

A workbook was completed over two weeks, allowing residents to respond to short-answer questions about their resource uses and habits (an example of a short answer question is: Do you have difficulties in getting to places?) along with 5- (5-scale Likert question example: How comfortable are you finding the house in relation to temperature? Very comfortable, mostly comfortable, neutral, mostly uncomfortable or very uncomfortable?) and 7- (7-scale Likert question example: How often do you use the public outdoor areas in WGV? Every day, a few times a week, about once a week, a few times a month, once a month, less than once a month or never?) point Likert scale survey questions. Text probes were sent periodically through these two weeks to gain in situ qualitative contextual data on current practices, minimizing the impact of recall difficulties during interviews [53]. The text probe method is a combination of cultural probe methods developed over the past two decades that requires

participants to take photos of objects during their daily life with a disposable camera [53–55]. The advent of mobile phones has allowed a significant advancement in this method. Text messages are a low-effort, quick and familiar method for the participant, increasing response rate. Examples of the questions used are, "Tell me how you have kept warm today?" or "In a picture or a few words, tell me what home means to you?"

Data analysis occurred after the first round of data collection and again after the second round. The Likert scale data was analyzed through tabular and graphical visualization of the results to identify trends, which were then compared with the qualitative data collected. A thematic analysis was performed using NVivo software to analyze the various data sources across 43 themes. (A short list of initial themes was drawn up before the thematic analysis based on the researchers' notes from the interviews, and this was then added to the analysis. Themes included affordability, comfort, control, convenience, energy, health, ownership, privacy, stability, thermal comfort, time, employment, cooking, fresh air, routine, washing, animals, children and sense of community.) It was during this analysis that the themes of home, sense of place and the concerns around moving to an LCD were identified as noteworthy. This paper is based on the further thematic analysis of the data with these themes in mind, following the method set out in [56] as well as the post-occupancy evaluation of how residents are experiencing life in the LCD. Quantitative methods are not the focus in this paper, however some results from the Likert scale questionnaire are discussed due to their relevance.

## 4. Results

This research explores the pre- and post-occupancy factors surrounding residents' motivations, perceptions and expectations of living in the LCD and how their emotional landscape of home is affected. These are important concepts to understand to further the acceptance of LCDs in cities of the future. These concepts were identified from the participants' answers to question in the interviews about how they heard of WGV, their motivations for moving in and their expectations for how their life would change (or not). These themes were then revisited in the post-occupancy interviews.

### 4.1. Resident Awareness of the Possibility to Move to a Low-Carbon Development

There are multiple ways that the residents became aware of WGV. These are shown in Table 2. Most residents discovered the LCD through their friends, some of whom were from a community group they were in (such as the SHAC artists community). Other residents had friends involved in other LCD projects in Western Australia that they were considering moving into, but then chose WGV instead. One resident saw news stories on local television regarding the development, while another two attended a local council event where it was mentioned. A number of residents heard about the development through work associates, either those who were moving into the LCD, or those involved in the development of the LCD. The strong influence of social networks in distributing information throughout the community was shown through the majority of residents discovering the LCD through personal connections.

**Table 2.** Fourteen residents' responses to how they originally heard about WGV.

| How Residents Heard about the Low-Carbon Development | Residents |
|---|---|
| Friend | 6 |
| Workmate | 2 |
| Local council event | 2 |
| Advertisements at LCD | 2 |
| TV media | 1 |
| Friend living in the LCD | 1 |

*4.2. Motivation to Move to a Low-Carbon Development*

Three main motivations for moving to an LCD have been identified through this research. These revolve around the design features of the LCD and the homes, the community aspects of the LCD and housing stability and control over space. As shown in Table 3, residents' motivations for moving to WGV were primarily due to the sustainability features of the homes and the development. This was followed by the attraction to living in a community of medium-density dwellings, as well as being able to interact with neighbors and engage in community events. For half of the studied residents, the LCD provided housing stability for them either by allowing them to purchase their own home or allowing them to lease an apartment belonging to SHAC, and the interviews uncovered a further aspect of this motivator: control over space. The final attractions of the LCD were the location, the design of the LCD and the dwelling design.

**Table 3.** Motivations of 14 residents for moving into the low-carbon development.

| Motivation for Moving into the Low-Carbon Development | Residents |
|:---:|:---:|
| Sustainability features | 10 |
| Community focus | 8 |
| Housing stability | 7 |
| Location | 3 |
| Dwelling size and attributes | 3 |
| Ecology of the LCD | 3 |

4.2.1. Attractiveness of Elements of the WGV Precinct

In a limited pre-purchase survey of some residents of WGV, 80% reported that the environmental sustainability features were a motivator to purchasing, while 100% reported the community attributes as a motivator. When asked how important these features were to them, all respondents reported them as very important or critical to their purchase decision. In the broader survey of the residents, as highlighted in the methodology, 10 out of 14 reported sustainability features as one of the motivations for moving to the LCD. The design of the dwellings and of the LCD were motivators also, at 3 out of 14 each. Residents believed that the system the LCD would create would enable them to perform the daily practices that they wanted to engage in and would bring satisfaction to their lives. This often revolved around the sustainable technology incorporated in the development, including the solar PV panels, rainwater tanks, community bore for garden irrigation and passive solar design features. For residents that previously had sustainability features in their homes, having these features in the LCD was an important motivator for the move. This highlights the acceptance and appreciation of sustainability features in housing, the economic and environmental benefits that were recognized by residents and the desire for these elements to be included in future homes.

The location of the LCD rated at 21% motivation for residents. For the residents of SHAC, the close urban center of Fremantle has traditionally been the artistic hub of the Perth metropolitan region and provided these individuals affordable and accessible housing and work spaces. Recently, local artists have been priced out of the housing market, and now have to travel long distances to reach their work spaces, exhibit their art or engage in the artistic community. The location was also popular with residents of other dwellings at WGV, due to the close proximity to farmers markets, preferred grocery shops, entertainment and social venues in Fremantle and, for some, proximity to work. Residents who purchased dwellings in Evermore and the single houses lived closer to the LCD pre-occupancy than those who moved into SHAC, as shown in Table 4. For residents moving into SHAC (the low-income subsidized housing for artists), they on average moved 16 km to live in WGV, with the greatest distance being 50 km. Those moving into Evermore and the houses were already closely located to the LCD and chose to stay in the same area, with distances of 8 and 1 km, respectively.

**Table 4.** Residents' pre-occupancy dwelling location distances from the LCD. Distances are in km and houses are grouped according to the development location post-occupancy in the LCD. House M has been left out due to the resident living between multiple houses each week, before moving to a stable house in WGV.

| Dwelling in WGV | House | Pre-Occupancy Dwelling Location Distance from LCD (km) | Dwelling Average Distance from LCD (km) |
|---|---|---|---|
| Evermore Apartments | A | 28 | 8.18 |
| | B | 1.8 | |
| | C | 4.3 | |
| | I | 5 | |
| | O | 1.8 | |
| SHAC Apartments | D | 0.6 | 16.32 |
| | H | 6 | |
| | J | 5 | |
| | L | 50 | |
| | N | 20 | |
| Semi-Detached House | F | 1.5 | 1.35 |
| Detached House | G | 1.2 | |

The design of the WGV precinct and the various dwellings was another motivator for residents to choose to live there. Residents of SHAC reported the industrial elements of the design as being reminiscent of the features along the port of Fremantle. The design also allowed them to make use of the multi-story steel walkways to showcase their artwork, and provided them with many spaces to congregate and socialize. For those who had lived in medium-density dwellings before, the return to this style of dwelling was an attraction (residents N and A). Other residents were attracted by the gardens and community space (resident I) that had been incorporated into the LCD.

4.2.2. Community Focus

Almost all residents discussed the community focus as part of their motivation for moving to WGV, either when specifically discussing why they moved to WGV or when discussing how they were expecting their lives to change. Residents discussed having more community connection with neighbors because of common interests, more community events or group projects between the groups (see quote below, Resident G) and more sharing of information (Resident O).

"The new lifestyle. Being part of something ... being part of a community" (Resident G)

Residents were excited about living somewhere with a specific design focus on the ambiance of community and feeling free to walk out their door without being forced to talk to someone they do not want to talk with (Resident J). The mixed ages of the residents was also appealing, allowing retirees to interact with children and different backgrounds to come together (Resident C).

For the SHAC residents, the motivator of living with people working in the arts or creative industries was a primary reason for moving to WGV. The expectation was that they would work collaboratively, either in the dedicated studio workspace or in the common green space of the SHAC dwelling. A year after moving in, there were already a number of community functions in the studio space, with residents working on creative projects together and children frequently seen playing freely between the apartments.

### 4.2.3. Housing Stability and Control over Space

For the residents moving into the affordable housing provided by SHAC, one of the main motivators for moving to WGV was stable, affordable housing. All SHAC residents had been renting or living with family and receiving subsidized rental assistance from the federal government. Resident N was particularly excited about living in a stable house after being homeless and living with friends for many years, as highlighted by this quote:

" . . . there's this beautiful . . . exciting development happening . . . you could well end up with somewhere to live." (Comment made to Resident N about the LCD development.)

For most SHAC residents, the sustainability features of the LCD were not a primary motivator, although the financial benefits of sustainable housing design and the inclusion of energy- and water-efficient aspects were attractive to them. This reflects the results found in [34,35], where stable housing was found to be one of the most important characteristics of a home that residents look for. For residents of the other cohorts, most did not mention housing stability as a motivator for moving into WGV. These residents had a combination of previous dwellings that they rented or owned. For a resident (M) who had been staying between friends' houses for the past three years, having stable housing of their own and not needing to move around frequently was a motivator for moving to the LCD.

Along with housing stability, having control over their own space without the intervention of a landlord was motivating for many residents. These responses all featured in the residents' motivations for moving to WGV. Being able to craft their homes to support their personal identities while living a sustainable life and participating in activities with like-minded people was a common motivator (resident C). Managing the home to facilitate their work was particularly important to the residents of SHAC, who had an additional space in their apartments that was a dedicated office and art space where they could work. Resident J focused on being able to arrange her home so the light was maximized for her artistic activities. For resident C, whose two teenage children were not moving to WGV with her, downsizing to a more manageable space was an important motivator. She had a different spatial environment compared to other participants, as she does not have to accommodate other people as often in her home practices [57]. A few residents discussed changes to keeping thermally cool or warm in the LCD. One resident (M), who had been living with friends for the past three years, was relieved to finally have complete control over the thermal comfort of their home to optimize it for maximum efficiency when performing different tasks. This resident liked the home to be cooler when working and warmer when relaxing.

### 4.3. The Perception of Life and Home in a Low-Carbon Development

While attraction to the physical attributes of the LCD is important, the reflection of what a home is in an LCD is also important to attract people in the future. A home in this paper refers to the meaning that residents ascribe to the physical building they inhabit [32]. The residents surveyed highlighted many aspects of a home that are important to them. Table 5 shows that the most frequent features are a sense of community, social aspects and family interactions. Not surprisingly, and given that these residents committed to moving into an LCD, environmental sustainability was the second most important feature of a home. The aesthetics and design of the home is the next important feature for residents. Despite significant literature emphasizing the security and safety aspect of housing, these results show that only 5 out of 14 residents considered this to be a desirable feature in a home. This was reported primarily by the SHAC residents who had not had stable housing in the past. The security and safety aspect of housing was rated below the design and aesthetic features of a home, which half the residents stated were important. In this research, comfort was desired by 5 out of 14 residents as an important feature in a home; however, the topic was raised in regards to thermal comfort many times during the interviews. This indicates that comfort is of importance to residents when involved with their daily practices, and so should still be considered a desirable feature in a home. These results are

similar to those shown in Table 3 (outlining the motivations for residents moving to the LCD), possibly indicating that they will feel satisfied with their move to WGV [16].

**Table 5.** The most important features of a home as reported by 14 residents.

| Important Features of a Home Pre-occupancy | Residents |
|---|---|
| Pets | 1 |
| Enjoyment, haven, relaxation | 1 |
| Garden | 3 |
| Location | 4 |
| Comfort | 5 |
| Safe, security, secure housing | 5 |
| Aesthetics, design of home | 7 |
| Sustainability | 7 |
| Community, social, family | 10 |

*4.4. Expectations of Living in a Low-Carbon Development*

Residents discussed their expectations of living in an LCD in the interviews structured around themes of sustainable living, sense of community and maintenance of their homes.

4.4.1. Living Sustainably

Many residents who had not previously lived in houses with sustainability features mentioned that they expected to be able to live more sustainability when they moved into the LCD. Residents used terms such as easier, normal and being supported when discussing the expectation that their daily practices would become more sustainable in their use of resources, including energy, water and transport. Resident H stated that the support from the community and the general focus throughout the LCD on sustainability would result in his daily practices changing to become more sustainable. Resident D made the statement that simply being in the LCD with a community of like-minded individuals would motivate her to change her practices and take environmental choices into consideration. She also stated that:

"Moving to a low carbon precinct I believe will improve my quality of life and motivate me to make better consumer and environmental choices." (Resident D)

Those residents who have already lived in a home with sustainability features did not expect many aspects of their life to change. They had expectations of travel practices changing, but discussions primarily revolved around the sense of community and opportunities to interact with their neighbors instead.

4.4.2. Low-Carbon Development Housing will be Easy to Maintain

A 2016 study investigated the ease of use of low-carbon technologies in the home [58]. It found that many residents had difficulty in using the technology, and recommend automation be considered to improve user perceptions. Other research has concluded that residents are concerned with the difficulty of maintaining their homes when sustainability features are employed [59]. The residents in this study, however, were confident in their abilities to maintain their LCDs, and believed that they would be able to perform the necessary practices required to maintain their LCDs (Table 6). There was a variety of answers from those residents who had sustainability features in their previous dwellings, and one responded neutrally, two were not concerned and one gave no response. Another resident (J) discussed having to change the way she warms the home, as she moved into a larger apartment in WGV with different design features and technology than she had pre-occupancy.

**Table 6.** Nine resident responses to the question "I am concerned about how easy it will be to maintain my low-carbon house" (five residents not included due to no response).

| *"I am Concerned about how Easy it will be to Maintain my Low-Carbon House"* | **Resident Responses** |
|---|---|
| Agree | 1 |
| Neutral | 2 |
| Disagree | 6 |

### 4.4.3. Concerns of Living in a Low-Carbon Development

Some concerns that are evident through interviews with residents have come from those who had not lived in a medium-density or LCD dwelling before. Individual residents were concerned with aspects outside of the dwelling design that they lacked influence over. This included neighbors' actions that may disturb them, the green space located within the LCD and how to incorporate it into their daily lives and the landscaping and management of the common garden areas. These concerns highlight the importance of communicating the benefits of living in an LCD effectively with residents, and clearly outlining policies and regulations before people decide to move in. Housing policies that are viewed as bureaucratic, or development regulations that are not well understood, may lead to conflict and produce a negative perception of the LCD as an attractive place to live.

### 4.5. Post-Occupancy Evaluation of Living in a Low-Carbon Development

In this longitudinal study, it is important to compare the experience of living in an LCD with a resident's expectations of life before they made the move. This will be discussed below, touching on the design aspects of the LCD, how the perception of the home changed and the community aspects.

### 4.5.1. Design Aspects

The design aspects and sustainability features of the LCD were major drawcards for residents choosing to make their home at WGV. Residents were attracted to the control they would have over their own space, the design of the apartments, the community space and the landscaping and green space.

The solar passive design features of the dwellings resulted in residents being more thermally comfortable in these dwellings than in their previous ones. Residents still have to engage in opening and closing doors and blinds, and putting on appropriate clothing; however, the space is reported to be more comfortable, and these actions are viewed as being more acceptable because the temperature remains in a more comfortable range. The design aspects also resulted in reduced energy usage by residents, adding to their attractiveness.

In terms of control over space, residents reported enjoying making their homes the way they wanted, and felt a sense of pride and ownership in their homes, with hope for future housing stability. This was particularly evident in the SHAC residents, who were motivated by this for the move. The community aspects will be discussed in a following section; however, one design aspect that was supposed to increase community interaction was the communal barbecues. These are placed in some green space in the north-west corner of the development. Their location was a deterrent for residents whose homes were located away from this area. The Evermore development also has their own communal barbecues inside their development, which were preferred by some residents due to their proximity. Other Evermore residents reported a preference for the WGV communal barbecues due to the surrounding landscape providing seating and shade and the communal atmosphere of interacting with other residents walking by.

A design aspect of Evermore that was criticized by all WGV residents was the extra lighting at night around the buildings and carpark. While this is intended for security reasons, and has been dimmed slightly in response to resident complaints, many WGV residents had to shut their curtains at night to be able to sleep, preventing them from having windows open to passively cool their homes. The lighting and security gates and fencing around the Evermore entrance ha residents reporting

feeling uncomfortable entering the development, associating it with a jail, a medical institution or a sporting field, not a home. On the other hand, residents did appreciate the security aspect that this gating and lighting provides.

### 4.5.2. Perceptions of Home

As discussed in Section 4.3, establishing a new home was a primary motivation for residents moving to the LCD. Before moving into the LCD, the important features of a home included community aspects, sustainability features, design, security and comfort, and these aspects are compared to the post-occupancy responses in Figure 1. These features were mostly still expressed post-occupancy. The most common feature associated with home after the move was that it is a place that is a haven from the outside world, relaxing and soothing to be in. This was previously reported by only 7% of respondents, and now was reported by 57%. The next most important feature was the family and social aspects of the home, decreasing from 71% previously to 43% post-occupancy. This was rated at the same importance level as comfort aspects of the home, which were rated at 36% before the move. The safe and secure housing aspects were reported at 36% post-occupancy, the same as pre-occupancy. The design and aesthetics of the home decreased in importance from 50% reported before the move to 21% after. A new feature mentioned after the move was the ability for a home to be a place where the residents can be creative and express themselves. This was reported by one resident in each of the SHAC and Evermore developments. The garden and pets aspects of home both remained important features. Noticeably, the sustainability aspects of home were not mentioned since residents moved into the LCD, despite their importance before the move.

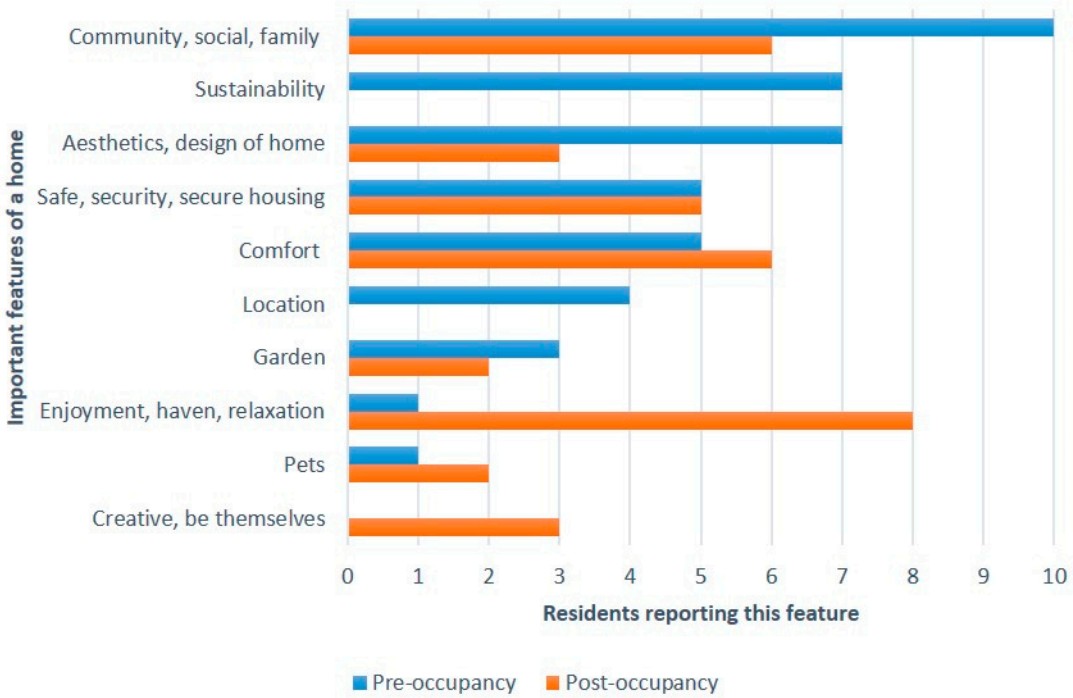

**Figure 1.** Important features of and associations with a home as reported by 14 residents pre- and post-occupancy in the LCD.

### 4.5.3. Community Aspects

The community focus was a driving force for many residents wanting to move into the LCD, with 57% of residents reporting it as being a motivating factor. Since moving in, all the residents reported enjoying the community atmosphere of the LCD. This included both those in the apartment buildings (SHAC and Evermore) and those in the semi-detached and stand-alone houses. Those in the apartments enjoyed the informal interactions with their neighbors as they walk about their development, while

those in the houses enjoyed being in their garden and interacting with residents walking past as well as the more formal gatherings organized within the LCD. Residents have reported enjoying having younger children around the apartment areas and listening to them play. A SHAC resident with a young child reported enjoying the community atmosphere of hearing other families cooking breakfast at the same time each morning, and this encouraged the resident to have a cooked breakfast themselves, instead of cereal, and to engage in the practice of preparing breakfast more mindfully.

Whilst the community aspects were highly praised by the residents, there were some aspects raised by residents relating to areas that had influenced their experience of community in the LCD.

The SHAC residents were some of the first to move in to the LCD, and some reported feeling like they had a responsibility to present a profile of a good community to the surrounding community outside of the LCD, even when their pets or kids did something that others did not believe to be socially acceptable. SHAC holds regular community events and workshops, and encourages the broader WGV community and residents of surrounding areas to attend these.

One resident of a single house felt that a greater effort to have had the residents of the single houses move in at a similar time would have allowed for more of a community to develop, and would have resulted in less construction noise impacting residents. Currently, that resident is waiting for both her neighbors to complete their homes before she completes her outside landscaping, and this is affecting her association with and comfort at home.

A community hub was mentioned by residents as a gathering space for those who do not otherwise have a space (such as the Evermore and SHAC common areas), or simply as a space for all WGV residents to interact regardless of their dwelling location. There is a building on site that could be used for this purpose when not being used for other community events such as workshops and dance classes; however, the organization of the hub or events would need to be done by either a resident or an external actor.

Finally, there was the realization by some residents that not everyone moving into WGV was embracing the community living or sustainability aspects of the site, and some residents just wanted to live as they would in more traditional housing developments, without engaging in these features or events. As the community and sustainability aspects were reported by the residents studied to be major motivating factors for the move to the LCD, this is not surprising; however, some residents stated that this was a surprise, and they had to respect the decisions of other residents.

## 5. Discussion and Conclusions

This paper set out to explore residents' motivations and expectations for how their life would change when they moved into an LCD, and the post-occupancy evaluation of this. The concept of home in the literature revolves around technical perspectives (technology features), social perspectives (comfort, social place, physical use) and sustainable practice (sustainable housing) perspectives of home [30]. The aspect of comfort in social housing policy generally relates to thermal or physical comfort, something that can be measured, predicted and changed through design adaptations [31]. Previous research has found that the future of comfort remains fluid and controversial [60]. Some of the results in this paper support this view, where the traditional notions of home design only focus on thermal comfort. In taking a social view of home perceptions and expectations of a move to an LCD, however, this research highlights the varying results that occur when non-technical aspects of an LCD are considered [30]. Home in this paper has been outlined to be primarily a place for community, sustainability, aesthetic features, safety and comfort, although the sustainability aspect reduced in importance once residents moved into the LCD. This could be due to residents easily integrating their practices and technology in the new environment and focusing more on the community aspects of their lifestyles. The different meanings of home revealed through this research point to various opportunities and obstacles for reducing resource consumption in homes [61]. Future research should focus on how the meaning of home influences individual and household resource consumption, and

investigate how living in an LCD impacts these. This could then inform more appropriate policy making related to homes and resource use that does not solely focus on the built environment.

In the literature, a primary aspect of the meaning of home is the importance of control over space, whether in relation to personal identity, security, comfort, privacy or activities [32,39,40,46]. These results are the same even for households with sustainability features [62,63]. This is supported in this research, as residents were particularly motivated by housing stability and having control over their own space [34,35]. Other motivation results also reflect the conclusions made by previous research [34], that the external environment of the community is important to people in a home along with housing stability. Location and design are common factors in purchasing a home anywhere, let alone in an LCD, and are replicated in this study as common features people look for in a potential home [16].

Previous studies have shown a strong desire from residents to have sustainability features in their homes, and the WGV precinct provides them with this opportunity [64]. The sustainability features of the LCD in this research were rated as a strong motivator for residents, followed by the community aspects being fostered at WGV. Residents believed that living in the LCD would enable them to develop practices that require less resources, increase their interaction with the community and change their travel practices. Residents of the LCD primarily found out about the opportunity to move into the LCD through their social networks of friends and workmates. Social networks are a trusted and familiar source of information for people in society, and hence might be used by real-estate agents to increase awareness of, and interest in, LCDs. LCDs feature design aspects and technology that require resident interaction to ensure their optimal performance. These can be of concern for prospective residents, as shown previously [20], although the residents in this study were not concerned about these features pre-occupancy. Designers, planners, real-estate agents and strata managers need to explain these clearly to prospective residents to ensure the technology is maintained in good working order to achieve the sustainability outcomes of the development.

The expectation of a strong sense of community pre-occupancy concurs with the findings from many studies on the important features of a home including the community aspects [34,35,41]. The strong sense of community and the self-reported thermally comfortable homes met residents' expectations post-occupancy, and are a positive selling point for future LCDs. Some design and community aspects were met with surprise in this research. The lighting and security aspects of the Evermore development received mostly negative views from the residents as influencing the ease at which they could move about the LCD precinct and interact with other residents. The communal barbecues also had mixed reactions, engaging some residents but not all. Other options for community interaction and meeting places should be explored to accommodate other preferences.

Research focusing on questions of the home often examines only the physical and techno-economic aspects of the built environment of the dwelling that people reside in. Those studies that focus on the home tend to include social and emotional connotations along with the built environment [20]. If policy is only focused on the built environment, then human social and emotional connections with their home may be neglected [24]. As these are important elements of social practices, any programs designed to influence resource use in the home are unlikely to result in long-term change. The emotional landscape of a home is increasingly being recognized as significant to residents, including in this study, and its relevance should be advocated for in housing policy, along with the physical structure of the dwelling [33,37,48].

For housing policy to lead to attractive homes in the future, it is important to understand which elements of the design of a home are desired by residents post-occupancy, and how these features influence daily practices. In terms of a policy approach towards housing, the WA Housing Authority acknowledges the desire for residents to have a safe, secure, stable house, and provide various dwelling types to meet residents' needs [62]. It is clear from this review that the term "home" is a complex system of physical and emotional elements [63], and the various ways of categorizing it provide opportunities to change resource consumption in related practices. It is with this open policy direction

in mind that this paper explored how residents perceive their homes and what they expect out of the LCDs that are being built to withstand future environmental climate change.

The authors acknowledge that this paper features results from a small cohort of LCD residents; however, it is unique in tracking them both pre- and post-occupancy. This was mostly due to the low uptake in residents who fit the time limit criteria for moving into the LCD in 2018. Some residents were also reluctant to participate, due to not having stable housing pre-occupancy, as this influenced the energy and water aspects of the research not discussed in this paper. However, with a small cohort study, particular themes could be examined in greater detail with the residents, such as how the different methods of hearing about the LCD influenced their decision to move in. Future research should examine a larger sample size of residents from different locations to assess whether other themes and concerns arise. A second post-occupancy study could also be completed once residents have resided in the LCD for a longer period of time. For most of the residents in the stand-alone and semi-detached houses and Evermore, they began living at WGV less than six months from when this data was collected. The SHAC residents had been residing at WGV for more than a year. This may have influenced their perceptions of their experiences.

Further research areas should continue to investigate LCD housing in a variety of climatic and design landscapes outside of the Australian and European regions to broaden the lessons learnt, the residents engaged with and the policies that affect LCDs. Mixed method research focusing on a longitudinal view of LCD residents is vital for understanding how residents access an LCD, move in and settle over the years with new technology and communities. Post-occupancy evaluation studies will contribute to this understanding.

**Author Contributions:** Conceptualization, J.K.B.; Formal analysis, J.K.B.; Funding acquisition, J.J.B. and G.M.M.; Investigation, J.K.B.; Methodology, J.K.B. and G.M.M.; Project administration, J.J.B. and G.M.M.; Resources, J.J.B. and G.M.M.; Supervision, J.J.B. and G.M.M.; Visualization, J.K.B.; Writing—original draft, J.K.B.; Writing—review & editing, J.K.B., J.J.B. and G.M.M.

**Funding:** This research is funded by the CRC for Low Carbon Living Ltd. supported by the Cooperative Research Centres program, an Australian Government initiative, NP2006.

**Acknowledgments:** The authors wish to especially thank the research participants from the WGV precinct for participating in this research and all the stakeholders involved, particularly LandCorp and the City of Fremantle.

**Conflicts of Interest:** The authors declare no conflict of interest.

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
