# Peer review of "Pre- and Post-Occupancy Evaluation of Resident Motivations for and Experiences of Establishing a Home in a Low-Carbon Development"

_sustainability, doi:10.3390/su11143970_

Round 1

Reviewer 1 Report

The paper is a valuable contribution to the discourse on low energy housing. The theoretical framing of the research is clear. Minor outstanding questions are as follows:

p.3 Methods section: A diagram or table summarising the timeline of research tasks and involvement of participants might facilitate reader’s understanding of research design.

Fig.1 Given there were 14 respondents presenting the frequency as “%” raises questions: how come some answers attracted 14% and other 15% of the vote if one respondent is 7,…% of the population. Similar question applies to Fig.2 and 3. For a question presented in  Fig.4 less responses were received (9) so the percent/per respondent is different. I’m not fully convinced such small sample size is well presented as percentage instead of absolute numbers.

In the Results section, not only results are introduced but are discussed in the context of findings from other studies (e.g. p.5 l.171). Perhaps these could be moved to the discussion section?

p.13, l.500-502 Other research revealing this issue is, e.g. “User learning and emerging practices in relation to innovative technologies…”

Reviewer 2 Report

Thank you for the interesting article which draws together a critical issue between energy efficiency and "liveable cities".

I have some comments which could improve the paper prior to publication and I think care should be taken with the methodology. At present I find the exact form of the study difficult to follow and this leads through into the results section. These two sections could be drafted in parallel to improve the flow of the paper and the clarity of the findings.

General comments:

Section 1 - Introduction. I think more information could be provided on the case study. Section begins by immediately discussing Perth and I think some background context on Australia and Perth, and why they make an interesting case study should be included here.

Page 1, line 37 - "In Australia, building codes require houses to 37 meet minimum energy efficiency requirements" . some examples of what these might be would be useful for context

Section 3 - could more information be given on the sample? What special characteristics are associated with people who live in low carbon homes?

In the methodology section it is assumed there is a lot of prior knowledge about Australian regulations. Perhaps an explanation of what "7+ star NatHERS" is, at least as footnote

Please explain what "living laboratory mixed methods" means

I would like to see a lot more detail in the methodology

·         How was the sample developed?

·         Who was contacted, why and what was the response rate?

·         Can you include some question schedule?

·         How many points were on the likert scale?

·         How many questions were asked and of what nature?

·         Did anyone drop out during the study?

The themes for the coding should be listed and how they were determined. Were the themes decided in advance? Did any additional theme emerge during the analysis of the interviews?

I'm not sure if I missed anything but I don't understand what WGV is?

Section 4

The colour coding for the pie-chart is a bit confusing, the text is clear but glancing at the chart makes it difficult to distinguish between “friend” and “TV Media”

Due to the unclear coding categorisation it is difficult to follow where the information for the results has come from

Section 5:

The policy implications and future research are discussed briefly on p13 line 488 and then stated in detail later on. I think the brief discussion can be removed.

I would like to see some discussions about the limitation of the research process and what areas could be improved upon, how could this fit into the future work plans?

Round 2

Reviewer 2 Report

The authors have addressed my comments and I am satisfied with the changes made. I am happy to recommend the paper for publication.